# On the Nature of the Word-Reduction Phenomenon: The Contribution of Bilingualism

**DOI:** 10.3390/brainsci9110294

**Published:** 2019-10-27

**Authors:** Sara Rodriguez-Cuadrado, Cristina Baus, Albert Costa

**Affiliations:** 1Departamento Interfacultativo de Psicología Educativa y de la Educación, Universidad Autónoma de Madrid, 28049 Madrid, Spain; 2Departament de Tecnologies de Informació I les Comunicacions, Universitat Pompeu Fabra, 08002 Barcelona, Spain; cristina.baus@upf.edu (C.B.); albert.costa@upf.edu (A.C.); 3Institució Catalana de Recerca i Estudis Avançats, 08010 Barcelona, Spain

**Keywords:** word reduction, language switching, bilingualism, duration, intensity, pitch

## Abstract

Word reduction refers to how predictable words are shortened in features such as duration, intensity, or pitch. However, its origin is still unclear: Are words reduced because it is the second time that conceptual representations are activated, or because words are articulated twice? If word reduction is conceptually driven, it would be irrelevant whether the same referent is mentioned twice but using different words. However, if is articulatory, using different words for the same referent could prevent word reduction. In the present work, we use bilingualism to explore the conceptual or articulatory origin of word reduction in language production. Word reduction was compared in two conditions: a non-switch condition, where the two mentions of a referent were uttered in the same language, and a switch condition, where the referent was said in both languages. Dyads of participants completed collaborative maps in which words were uttered twice in Catalan or in Spanish, either repeating or switching the language between mentions. Words were equally reduced in duration, intensity, and pitch in non-switch and in switch conditions. Furthermore, the cognate status of words did not play any role. These findings support the theory that word reduction is conceptually driven.

## 1. Introduction

Word reduction shows that predictable words are likely to be shortened in a variety of ways, such as in their duration or intensity [1,2,3,4,5]. The predictability of words can come from different sources, such as their lexical frequency, sentence context, or previous mentions in a discourse; furthermore, word predictability can be manifested in shorter durations and lower intensities, or in a decrease in mean pitch [2,6,7,8,9,10,11,12,13]. Although word reduction has been widely described, little is known about its origin. As pointed out by Lam and Marian [14], models of speech production often identify a conceptual level, a lexical level, and a phonological level. The aim of the present work is to explore the nature of the word-reduction phenomenon by examining conceptual and articulatory influences. 

There are some studies that have provided indirect evidence to account for conceptual or articulatory influences in word reduction [4,15,16,17,18,19,20]. Fowler [17] found reduction when the same words were repeated, but not if using homophones (words which are identical in sound but different in meaning), which speaks in favor of reduction taking place at a higher level than phonological encoding [21]. Content and function words can also be informative about the origin of word reduction. Content words are defined as meaningful words, such as nouns, adjectives, verbs, or adverbs, whereas function words structure sentences, as made by pronouns, prepositions, articles and conjunctions. If word reduction is mainly driven by articulation, both content and function words should be reduced. However, if reduction is conceptually driven, content words should exhibit higher levels of reduction, as they convey more meaning than function words. Bell et al. [3] performed regression analyses on the Switchboard corpus [22] finding reduction in duration for content words, and not for function words. Similar results were obtained by Jurafsky et al. [4], agreeing with a conceptually driven source for reduction. Finally, word reduction has been found when the two repetitions of an item have different articulatory forms, for instance, when second mentions are anaphors of the target word [9,23]. 

On top of the reviewed evidence, it is also necessary to consider those proposals that, although not neglecting the relevance of concepts, argue that articulation leads to more reduction. The Facilitation-Based Reduction Hypothesis [19] assumes that speakers reduce words because speech production is facilitated. Other studies have explored whether self-repetitions (that is, when the same word is produced twice by the same person) versus cross-repetitions (the same word is produced twice but one by each person belonging to a pair) of a referent lead to different amounts of reduction. If self-repetitions were consistently more reduced, this would support the Facilitation-Based Reduction Hypothesis [19]; however, the evidence is mixed. Bard and Aylett [16] (see experiment 3) found that reduction was unaffected by who uttered the first mention. On the other hand, Trón [20] analyzed recordings from the Edinburgh Maptask Corpus and found that self-repetitions were more reduced than cross-repetitions.

Crucially, we believe that an interesting way of investigating the origin of word reduction is through bilingualism. Lam and Marian [14] approached this question with balanced and unbalanced English–Korean bilinguals (unbalanced bilinguals who were either English or Korean dominant). Participants engaged in a two-sentence event-description task, where the referent could be named either once or twice. Repetitions were uttered either in the same language (English) or by switching from Korean to English. Authors examined target words’ durations and intensities (both raw and relative to the whole utterance), finding mixed evidence regarding the conceptual or lexical origin of word reduction. Balanced bilinguals displayed reduction regardless of which language they used, which speaks in favor of conceptually driven word reduction. For unbalanced bilinguals, however, reduction seemed to be operating at the word level, being mostly present when the same language was used for both utterances. 

In the current study, early and balanced Spanish–Catalan participants completed a collaborative map task taken from Rodriguez-Cuadrado et al. [5], where items were presented twice, in order to elicit word reduction through repetition. To explore the nature of word reduction, the same referent could be named with two different words, either in Spanish or in Catalan (signaled by a Spanish or a Catalan flag). Also, in contrast to Lam and Marian [14], the switch was bidirectional (rather than unidirectional, from Korean to English): half of the time, participants switched from Spanish to Catalan, while the other half was from Catalan to Spanish.

The rationale for undertaking our study is that, if word reduction is mainly driven by the predictability of the referent, it would be irrelevant how the object is named if it is referred to twice. However, if word reduction is affected by the predictability of a word’s articulatory form, using two different words might impede word reduction. Additionally (in line with Lam and Marian [14]), there is also the possibility that both forces are at play. Thus, even if we were to find reduction when there is a language switch, this does not necessarily imply that only concepts are at work, but rather that their contribution could be larger than that of articulation. In order to further explore this issue, we manipulated the cognate status of the words used. Half of our stimuli were Spanish–Catalan cognate words (e.g., “piña” in Spanish and “pinya” in Catalan, meaning “pineapple”), and half were non-cognates (e.g., “botella” in Spanish and “ampolla” in Catalan, meaning “bottle”). Cognates can be defined as words belonging to different languages which share an etymological origin; thus, the criterion to decide whether word translations are cognates is based purely on phonological or orthographical similarities [24]. Many authors have reported a facilitatory role of cognates during speech production in bilinguals. Consequentially, we hope that the use of cognates will help us elucidate the role of phonology in word reduction [25,26,27,28,29,30,31]. Having said that, if reduction is conceptually driven, balanced bilinguals should reduce words to an equal extent in repeat and switch trials, whereas, within switch trials, there should be no differences between cognates and non-cognates. On the other hand, if reduction is phonological (being conceptually driven as well or not), it would be logical to find more reduction in repeat trials than in switch trials, while one would expect more reduction for cognates than for non-cognates within switch trials. 

As argued by Costa et al. [24], discussions about the origin of the cognate effect have located it at three potential levels: conceptual–semantic, lexical–morphological, and phonological–sublexical, although most accounts have favored a phonological account. Also, as Costa et al. [24] suggest, if we track cognate effects to shared semantic representations, that would ultimately mean that semantic representations of cognates are more similar than those of non-cognates. Furthermore, this would imply that the semantic representation of semantically related word pairs that are cognates in one language of a bilingual (e.g., “pen” and “pencil” in English) but not in the other (e.g., “bolígrafo” and “lápiz” in Spanish) would be stronger for English than for Spanish. Costa et al. [24] proposed two processes involving cross-linguistic activation to explain the cognate benefit. From a cascading viewpoint, phonology is activated not only in the selected language but also for its translation in the nonresponse language [32]. Therefore, facilitation when naming cognate pictures, for instance, would suggest lexical activation in the unintended language that would spread activation to phonology. Cognates would thus have a double source of activation, where this would not be the case for non-cognates, as there would be no phonological overlap. A second and compatible explanation by Costa et al. [24] is based on interaction and bidirectionality between phonology and lexical items: Activating phonological information would affect lexical selection, which then in turn would send back some activation to connected words in the lexical network. Therefore, naming a cognate in a certain language would activate its translation.

### Predictions for Current Research

We predict that repeated words will be shortened both when the two mentions of an item are uttered in the same language and when there is a language switch, due to the evidence indicating that word reduction should be composed of more than just articulatory priming [3,4,14,16,17,18]. Regarding whether word reduction would work in the same way for repeat and for switch trials, we expect some conceptually driven reduction for both trials, but acknowledge that it is also plausible that articulation further facilitates word reduction. According to the Facilitation-Based Reduction Hypothesis [19], the heavier the facilitation in speech production, the larger the reduction. Therefore, articulatory priming should enhance word reduction, resulting in larger reduction when both mentions are uttered in the same language compared to when there is a language switch. 

For cognates, even though less word reduction would be expected for second mentions in the switch condition, mentions involving cognates were expected to be more reduced (as it implies priming of articulatory forms) than mentions involving non-cognates. Finally, as the participants are balanced Spanish–Catalan bilinguals, no differences in word reduction are expected depending on the language (in addition, evidence coming from text reading shows that unbalanced bilinguals also reduce words in their second language, at least in duration [33]).

To sum up, the following is predicted:Both repeat and switch second mentions will be reduced in comparison to first mentions.Due to a plausible larger amount of facilitation, second mentions in the repeat condition are susceptible to be more reduced than second mentions in the switch condition.Word reduction in the switch condition might be benefited by the presence of cognates, compared to non-cognates.

## 2. Materials and Methods

### 2.1. Participants

Thirty-one balanced early Spanish–Catalan bilinguals (nineteen females, mean age: 26.51 years, SD: 8.22) took part in the experiment. All participants reported speaking Spanish and Catalan from an early age, received their education in the two languages, and felt equally proficient in both languages. They received a 7 euro compensation for their participation. All participants had normal or corrected-to-normal vision and none of the participants reported having any speech or hearing impairments. Participants were recruited from the Universitat Pompeu Fabra subject database, which involved filling in consent forms and providing ethical approval prior to the study. The authors declare that the study followed the guidelines set by the 1975 Declaration of Helsinki.

### 2.2. Materials and Procedure

In order to examine the duration, intensity, and pitch of first and second mentions in the repeat and switch conditions, we used the collaborative map task of Rodriguez-Cuadrado et al. [5]. A Spanish–Catalan participant acted as the speaker, and a Spanish–Catalan confederate as the listener. Participants were not informed that the listener was a confederate. The speaker was presented with a sequence of 6 “maps” (see Figure 1) and an initial practice map shown through DMDX [34] on a computer screen. Each map contained 8 objects, distributed as follows: there were 2 arrays of 4 objects, 1 array displayed in a string in the upper part of the map, and 1 array in the lower part. Two objects at each time were linked in 8 consecutive steps per map. Each object of the map appeared in 2 different links, which could be horizontal (2 objects in the same string in the upper or in the lower part of the screen), vertical (2 objects in the same axis in different strings from the upper to the lower or from the lower to the upper part of the screen), or diagonal (2 objects in different axis in different strings from the upper to the lower or from the lower to the upper part of the screen). Additionally, in order to signal the language of instruction, either a Spanish or a Catalan flag was depicted at the top left of each map (see Figure 1).

Items consisted of 48 black-and-white line drawings selected from several sources (including the Snodgrass database [35] and the International Picture Naming Project [36]). Drawings corresponded to 48 Spanish and 48 Catalan words equated in number of letters and logarithmic frequency. Frequency values for both Spanish and Catalan words were obtained through the NIM software [37]. Half of the items were cognates, where Spanish cognates and non-cognates and Catalan cognates and non-cognates did not differ in terms of logarithmic frequency or number of letters (for a complete list of items and their lexical values, see Appendix A, Table A1). Words were randomly distributed regarding the following: (1) the map to which the item belonged, there being 6 options; (2) the order in which items were displayed in the maps’ arrays, there being 8 options; (3) the order in which they were mentioned per map, there being 2 options; and (4) the item with which words were paired, there being 7 options per map. Repetitions were not immediate through maps, but there were between 1 and 13 intermediate words between mentions (depending on the randomization in the mention order). Finally, half of the second mentions were uttered in the same language as the first mentions (repeat trials), whereas, in the other half, there was a language switch (switch trials).

Once randomization order and distribution of items were made, 4 different lists were created (for a detailed example of the lists per map, see Appendix A, Table A2 in English or Table A3 in Spanish and Catalan). Each list contained, for each map, 2 items whose second mention was a repetition in Spanish (e.g., “cuchillo” in Spanish (“knife”) and “cometa” in Spanish (“kite”) in List 1, Table A2 and Table A3 of the Appendix A), 2 items whose second mention was a repetition in Catalan (e.g., “xiulet” in Catalan (“whistle”) and “ocell” in Catalan (“bird”) in List 1, Table A2 and Table A3 of the Appendix A), 2 items whose second mention was a switch trial, uttering the first mention in Spanish and the second mention in Catalan (e.g., first mention “bolo” in Spanish, second mention “bitlla” in Catalan (“skittle”), and first mention “hucha” in Spanish and “guardiola” in Catalan (“piggy bank”) in List 1, Table A2 and Table A3 of the Appendix A), and 2 items whose second mention was a switch trial, being the first mention in Catalan and the second mention in Spanish (e.g., first mention “mico” in Catalan and “mono” in Spanish (“monkey”) and first mention “pinça” in Catalan and second mention “pinza” in Spanish (“peg”) in List 1, Table A2 and Table A3 of the Appendix A). Furthermore, it was ensured that those words that were presented at the beginning of the sentence of instruction (e.g., “skittle” in “go from the skittle to the piggy bank”) appeared at the end for half of the participants (thus, “skittle” in “go from the piggy bank to the skittle”). This resulted in a total of 8 lists. A total of 96 mentions were uttered per participant, with 48 as first mentions and 48 as second mentions. Out of the 48 first mentions, 24 were named in Spanish, and 24 were named in Catalan. The same distribution applies for second mentions. 

The listener had the same 6 maps (plus the practice map, which is available in Table A4 in the Appendix A) as the speaker printed on paper, with no links between the objects and without the language-cuing flag. The task of the speaker was to tell the listener what the two linked objects were, specifying their direction by uttering instructions, such as “go from the monkey (object 1) to the bottle (object 2)”, in the language cued by the flag. After each instruction, the listener would draw an arrow between the two mentioned objects. Each step in the map remained on the screen until the speaker pressed the spacebar after uttering the instruction. Participants were seated face-to-face in a soundproof booth, where the screen of the speaker’s laptop prevented both the speaker and the listener from seeing each other’s map. Utterances were recorded, labeled, and analyzed using Praat version 5.3.15 [38], obtaining duration (milliseconds), average intensity of the entire word (decibels), and average pitch of the entire word (hertz). Analyses of utterances were blind to the experimenter, so it was not possible to tell if the target word was a first or a second mention. The task lasted approximately 20 minutes.

## 3. Results

Results were analyzed with a multilevel linear mixed-effects model implemented in R (lme4 library [39,40]). Mention (1st mention, 2nd mention non-switch, and 2nd mention switch), cognate status (cognates and non-cognates), and the interaction between them were introduced as fixed effects, and the random-effects structure was defined by a forward model selection (starting from a model without random slopes). The model comparison did not show differences between the minimal model, including only the random effects of subjects and items (1/Subject + 1/Item), and a more complex model, including random effects and random slopes of each fixed factor (1+ Mention* Cognate /Subject + 1 Mention *Cognate /Item; χ2 = 34.6, *p* = 0.7). For simplicity, our effects were then based on a model with mention and cognate status as a fixed effects and subjects and items as random effects. The final model was based on 2179 observations (31 participants and 97 items). Cognate status variable was centered (allowing for a better interpretation of the interactions). In addition, mention was coded using Helmert contrasts (i.e., each level of a factor with the average of its subsequent levels), leading to two contrasts: the first Helmert contrast compared the mean duration/intensity/pitch of first mention to the second mention (regardless of the type of trial). The second Helmert contrast compared the second mention of non-switch trials to the second mention of the switch trials. Table 1 displays parameter estimates for the models, considering duration, intensity, and pitch. 

For duration, the model revealed an effect of mention in both Helmert contrasts. As indicated by the first Helmert contrast, durations were shorter the second time a word was mentioned (β = −20; SE = 4.7, t = −4.3, *p* < 0.001). Interestingly, the second Helmert contrast revealed that second mention durations were longer for switch than for non-switch trials (β = 15; SE = 6.7, t = 2.2, *p* < .05), indicating that duration reduction between mentions was more pronounced for those trials not involving a language switch (i.e., non-switch trials).

The main effect of cognate status was not significant (β = 16; SE = 14, t = 1.15, *p* =.2), revealing that durations were similar when naming cognates and non-cognates. Cognate status did not interact with mention (1st Helmert contrast; t < 1) or with 2nd mentions (2nd Helmert contrast; β = −11; SE = 10, t = −1.16, *p* = 0.2), showing that reduction between mentions was not modulated by the cognate status of the words (see Figure A1 in the Appendix A).

For intensity, the model revealed only an effect of mention. As for duration, repeated items had lower intensities than first-mention items (1st Helmert contrast: β = −0.7; SE = 0.1, t = −5.04, *p* < 0.001). Neither the comparison between second mentions, (t < 1) nor the interaction with cognate status revealed any significant effect (t < 1). 

For pitch, only an effect of mention was observed (1st Helmert contrast: β = −4.8; SE = 1.7, t = −2.7, *p* < .01). Repeated items had a lower intensity than first-mention items. None of the remaining comparisons were significant (all ts < 1). 

## 4. Discussion

The aim of the current work was to explore the locus of the word-reduction phenomenon. Bilingualism was taken as a means to ask whether word reduction is mainly driven by the repetition of meaning (activation of the same concept twice) or by the articulation of words (activation of the same phonology/articulatory plan twice). Dyads of early balanced bilinguals (one participant and one confederate) engaged in a collaborative map task in which 48 items were mentioned twice to assess word-reduction effects by repetition in duration, intensity, and pitch. Crucially, second mentions were uttered either in the same or in a different language than first mentions, so word reduction could be elicited within languages (repeat trials) or between languages (switch trials). Also, half of the items were Spanish–Catalan cognates. Within our rationale, if word reduction was conceptually driven, it would be irrelevant whether the second mention of an item was uttered in the same or in a different language than the first mention, as the concept is evoked twice. On the contrary, if word reduction was a phenomenon of a mostly articulatory nature, it would be more likely to be observed when both mentions were uttered in the same language than when each mention was uttered in a different language, due to reasons such as phonological/articulatory facilitation. Additionally, as cognates are similar from an articulatory point of view, more reduction would be expected in those switch mentions having cognates compared to those with non-cognates.

Three predictions were formulated in the introduction. First, we hypothesized that both repeat and switch second mentions would be reduced in comparison to first mentions. Our results supported this prediction for duration, intensity, and pitch, replicating previous findings in which speakers reduced words in many ways when involved in a collaborative task with a partner [13,15,16,41]. 

Our second prediction was that second mentions in the repeat condition would be more susceptible to be reduced than second mentions in the switch condition, due to a larger amount of facilitation. For duration, there was a reduction in both switch and non-switch trials, being more pronounced for non-switch trials, while word intensity and pitch were reduced in comparable amounts for both conditions. These results suggest that word reduction is largely conceptually motivated, but acknowledges a small contribution from articulation in word-duration reduction. Therefore, only in the case of word duration is the Facilitation-Based Reduction Hypothesis [19] partially supported, as it assumes that the speaker’s experience with the word facilitates pronunciation to a greater degree than the activation of the concept.

Our third prediction was that, if articulation facilitates word reduction, cognates in the switch condition should be more reduced than non-cognates. Results, however, did not follow this prediction, as reduction in switch trials was present regardless of whether the words, either in Catalan or in Spanish, were cognates or non-cognates (also, not finding a cognate effect in switch trials would not fit with the Facilitation-Based Reduction Hypothesis [19]). This evidence also backs up the larger implication of concepts in word reduction because, if articulation were fundamental for word reduction, switch mentions, including cognates, should have been more reduced than those without cognates. Therefore, without completely denying the additive role of articulation, we believe that concepts are the driving force of word reduction. 

The results of the current article suggest that duration is more sensitive to language switching than intensity or pitch (where we find no indication of switching affecting reduction). This pattern of results is in line with previous accounts claiming that, although duration, intensity, and pitch are susceptible to be reduced by repetition, they might be affected in different ways. Concretely, Lam [42] suggested that word duration could be more influenced by articulation than intensity, which fits the pattern of reduction observed in this study, since word duration reduction was stronger in non-switch trials. Along these lines, in the study of Lam and Marian [14], different patterns for raw duration and raw intensity were found. Namely, word duration was only reduced in the non-switch condition, but intensity was reduced whether or not there was a language switch. Even though, in our study, word duration reduction was also found in the switch condition, it was smaller than for the non-switch condition. Therefore, it seems that reduction for duration is more sensitive to articulation than it is for word intensity (see also [13]).

This study also provides evidence to the relevant field of bilingual speech production. In particular, our results extend previous observations of language-switching costs in naming speed [43,44] to word duration. When comparing second mentions, differences were observed between non-switch and switch trials. Importantly, the language-switching cost in word duration, together with the fact that second mentions were always reduced regardless of type of trial, suggests that word reduction is mostly a conceptually driven phenomenon but can be modulated by subsequent computations in the course of naming. With the present data, we cannot determine the post-conceptual level (e.g., lexical and articulatory) at which word-reduction effects are modulated. Within models of bilingual language production, differences between switch and non-switch trials might stem from cross-language interference at the lexical level between translation equivalents [43,44]. Note, however, that the same result can be explained without invoking lexical interference, but rather post-lexical processes during which repeated articulation might have resulted in a larger word duration reduction for non-switch trials than for switch trials.

One potential limitation in our study was the use of confederates. Spanish–Catalan confederates were chosen, instead of naive participants, mainly to reduce variability from the side of the listener, as we focused on the speaker’s behavior. The use of confederates is conventional when employing collaborative tasks like our map task or similar tasks [5,45,46]. A revealing example of the impact of using confederates comes from the study of Brown and Dell [47] and its posterior replication by Lockridge and Brennan [48]. In the study, speakers described scenes that contained typical or atypical instruments to a listener who either did or did not receive pictorial support for the scenes. If the listeners had pictorial support, speakers would not need to mention or describe the atypical instruments too explicitly. If, on the contrary, listeners did not have pictorial support, speakers would have to highlight the use of atypical instruments so that listeners could follow the story. In terms of results, the difference between Brown and Dell [47] and Lockridge and Brennan [48] was that, when speakers interacted with naive listeners as opposed to confederates, they gave descriptions that were richer and more adjusted to the listener. Therefore, although we believe that, for our specific research purposes, the use of confederates was more appropriate, we would like to stress that researchers need to be mindful of these potential differences in speakers’ behavior. Considering this example, one might wonder whether it is possible that naive participants could have behaved differently with our confederates. We believe, however, that, due to the simplicity of our task and to the reduced role that the confederate had (who virtually just linked two objects in paper maps), the impact of using naive versus confederate listeners in our specific case would be rather small.

To conclude, the current study suggests that word reduction is mainly and primarily modulated by conceptual influences, as speakers reduced second production of words even when they were uttered in different languages (that is, in switch trials) and irrespective of the order (Catalan to Spanish or Spanish to Catalan). Thus, this study directly supports, for the first time, that word reduction is mainly conceptually driven.

## Figures and Tables

**Figure 1 brainsci-09-00294-f001:**
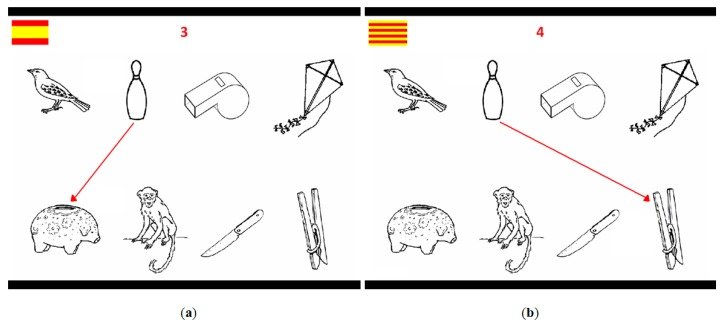
Example of the map task: (**a**) speaker in step 3 of the map utters “ve del bolo a la hucha” in Spanish (“go from the skittle to the piggy bank”); (**b**) Speaker in step 4 of the map utters “vés de la bitlla a la pinça” in Catalan (“go from the skittle to the peg”). The repetition of “skittle” from step 3 (to be named in Spanish) to step 4 (to be named in Catalan) constitutes a switch trial.

**Table 1 brainsci-09-00294-t001:** Mixed model estimates for word duration, intensity, and pitch.

	DURATION			INTENSITY			PITCH		
**Fixed effects**	**EST**	**SE**	***t*-value**	**EST**	**SE**	***t*-value**	**EST**	**SE**	***t*-value**
(Intercept)	468.5	12.9	36	68.4	0.5	120	178	7.8	22
Mention1 (M1) vs. Mention2 (M2)	−22.9	3.7	−6.1	−0.7	0.1	−5	−4.8	1.7	−2
Mention NS (MNS) vs. Mention SW (MSW)	11.9	5.3	2.2	0.06	0.2	0.3	1.2	2.4	0.5
Cognate (Cogn)	13.1	14.4	0.9	0.3	0.2	1.3	2.1	1.8	1.1
M1 vs. M2: Cogn	1.3	7.5	0.1	−0.02	0.2	−0.08	−0.6	2.6	−0.2
MNS vs. MSW: Cogn	−17.4	10.7	−1.6	0.1	0.4	0.3	3.2	3.8	0.8
**Random effects**	**VAR**	**SD**		**VAR**	**SD**				
Items	5099	71.4		0.68	0.8		29.8	5.4	
Subjects	3421	58.4		9.6	3.1		1881	43.3	
Residual	7421	86.15		11.6	3.4		954	30.8	

EST, estimate; SE, standard error; VAR, variance; and SD, standard deviation.

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
