# Peer review of "On the Nature of the Word-Reduction Phenomenon: The Contribution of Bilingualism"

_brainsci, 2019, doi:10.3390/brainsci9110294_

Round 1

Reviewer 1 Report

Summary:

The study reports on a behavioural experiment that investigates the location of the word reduction effect for repeated words (conceptual or articulation level). Spanish-Catalan bilinguals were tested on a collaborative map task, in which a speaker had to describe the connection between two line drawings. Repeated stimuli could involve a language switch, and could be cognate or non-cognate items. The results point to an effect of reduction for repeated items regardless of language switching or cognate status, suggesting that the reduction effect is of a conceptual nature.

The manuscript reports on a very interesting, well-designed and properly controlled study, the rationale of which is sound and the results are clear, such that my overall impression of this submission is positive. Given the behavioural nature of the experiment, it is not entirely clear to what extent this is of interest to the readers of a strictly neuro-scientific journal. Other than that, I have a few comments that pertain to the possible occurrence of a language switch cost, and some methodological aspects that can be clarified further.

Main comments

Firstly, given the inclusion of language switches in the design of this study, I was curious why the authors do not discuss the possibility of language switch costs at any point in their manuscript. The occurrence of switch costs is ubiquitous, even for the group of balanced bilinguals that are under scrutiny here (e.g., Costa & Santesteban, 2004). With word repetitions that are spaced apart by 1-13 intermediate words, I would expect see effects on the naming times. I understand this is not what the authors are after; instead they are interested in the effects on the variables of duration, intensity and pitch, but the pattern observed for the duration results made me wonder if this could not be explained by switch costs. Duration reductions were more pronounced for non-switch trials (l.236-7), which could suggest that switch trials were processed slower. The authors explain this by saying that a small contribution from articulation may be involved (l.277), but I would like to know if this possible articulatory effect could in some way be related to switch costs. Perhaps more generally, the discussion does not elaborately address how these new findings relate to previous word reduction findings (it mostly focusses on the role of the confederate). So, could the authors discuss in more detail how the results relate to previous findings, and what the role of switch costs may be? As a side note, in this regard I would also be interested to know if the authors have tried running a model that included language as a factor.

Secondly, in their predictions for cognates, the authors only seem to consider a possible difference between cognates and non-cognates to be evidence for priming of articulatory forms (l.129). Yet, prior to that, they state that semantic representations of cognates may be more similar that those of non-cognates (l.104). Would an effect of cognates, in line with previous work on homophones (ll38-41), thus not also in part be interpretable as a conceptual effect?

Thirdly, the authors should explain in more detail how their measured their dependent variables. Duration is clear, but at which point in the word were intensity and pitch measured? Where these averages over the entire word duration, or measured at a single point?

Minor comments

The manuscript contains some descriptions that were not entirely clear to me (in some cases this may have to do with language issues). This concerned the following statements:

78 “where first mentions in Lam and Marian’s study were assessed globally” (note double spacing after Marian’s). Does this refer to an average of all the first mentions of a referent? 86 “that does not necessarily imply that only concepts are at work, but that their contribution is larger” What exactly does the larger contribution refer to? Of what, or to what? 106-7 “this would imply that the semantic representation of those objects being cognates in two languages but non-cognates in a third language is stronger for cognates” It seems like this sentence is incomplete, or I misunderstood the last part: what exactly is stronger for which type of cognates? Also note the extra bracket after “cognates).” 120-123 “Regarding… reduction” Please check the sentence (two subordinate clauses before the main clause are hard to read)

Other comments

147: missing word after filling (in) 147: missing article before Authors 151: reference 15 does not seem to be correct (Bard et al, instead of Rodriguez-Cuadrado et al) 154: “showed” should be “shown” 188-191 Repeated use of “being the first mention” seems incorrect 249: word missing in “resulted significant” 267: Four should be three 286: change “as” to “because” and “was” to “were”(subjunctive) 292: “The use of confederates is very extended” does not sound right to me 299: “for a correct following of the story” is not correct The discussion of the confederates made me wonder if actual participants were informed that they were interacting with a confederate rather than a naïve participant. Please report this in the Method.

Finally, I am sorry for your loss, and I will miss Albert’s presence at conferences.

Reviewer 2 Report

Excellent paper on the locus of acoustic reductions, which suggests that reductions are primarily caused by repetition at the conceptual instead of the phonological level. By applying a paradigm from bilingualism research to study reduction, these researchers have been able to shed more light onto the underlying cause of the reduction phenomenon.

General comment:

Results suggest that speakers durationally reduce to a larger extent in same-language repetitions compared to between-language repetitions, whereas intensity/pitch reductions are similar within and between languages. How do you explain this finding? Could you speculate on why this difference in reduction within/between languages can only be observed in terms of duration? 

Minor comments:

Line 71: operating at word level -> operating at the word level

Line 78: Marian's study (spacing issue)

Line 78: where assessed globally -> were assessed globally

Line 107: Catalan)" -> Catalan)

Line 109: on the selected language but for its -> in the selected language but also for its

Line 239-240: Cognate status did not interact with Mention (1st Helmert contrast; t < 1) neither with second mentions -> Cognate status did not interact with Mention (1st Helmert contrast; t < 1) or second mention

Line 243: For intensity, the model revealed only an effect of mention -> For intensity, the model revealed only an effect of Mention

Line 244: lower intensity than first -> lower intensities than

Line 248: had lower intensity -> had lower intensities

Line 248-249: None of the remaining comparison resulted significant, (all ts<1) -> None of the remaining comparisons were significant (all ts<1).

Line 284: (also, not finding a cognate effect in switch trials would not fit with the Facilitation-Based Reduction Hypothesis [19]) -> Confusing -> (also, not finding a cognate effect in switch trials does not fit the Facilitation-Based Reduction Hypothesis [19])

Line 291: naïve -> naive (also in other places)

Line 292: The use of confederates is very extended -> The use of confederates is conventional

Line 296: who either had, or had not, pictorial support for the scenes. -> who either did, or did not, receive pictorial support for the scenes.

Line 312: Thus, way, this study directly elucidates, for the first time, that word reduction is mainly conceptually- driven. -> Thus, this study directly elucidates, for the first time, that word reduction is mainly conceptually- driven.
